# Disease-Specific Quality Indicators for Outpatient Antibiotic Prescribing for Respiratory Infections (ESAC Quality Indicators) Applied to Point Prevalence Audit Surveys in General Practices in 13 European Countries

**DOI:** 10.3390/antibiotics12030572

**Published:** 2023-03-14

**Authors:** Akke Vellinga, Addiena Luke-Currier, Nathaly Garzón-Orjuela, Rune Aabenhus, Marilena Anastasaki, Anca Balan, Femke Böhmer, Valerija Bralić Lang, Slawomir Chlabicz, Samuel Coenen, Ana García-Sangenís, Anna Kowalczyk, Lile Malania, Angela Tomacinschii, Sanne R. van der Linde, Emily Bongard, Christopher C. Butler, Herman Goossens, Alike W. van der Velden

**Affiliations:** 1School of Public Health, Physiotherapy and Sports Science, University College Dublin, D04 V1W8 Dublin, Ireland; 2Research Unit for General Practice, Department of Public Health, University of Copenhagen, DK-2200 Copenhagen, Denmark; 3Department of Social Medicine, School of Medicine, University of Crete, 71003 Heraklion, Greece; 4Balan Medfam Srl, 400064 Cluj Napoca, Romania; 5Institute of General Practice, Rostock University Medical Center, 18057 Rostock, Germany; 6Department of Family Medicine, “Andrija Stampar” School of Public Health, School of Medicine, University of Zagreb, 10020 Zagreb, Croatia; 7Department of Family Medicine, Medical University of Bialystok, 15-089 Bialystok, Poland; 8Department of Family Medicine & Population Health, University of Antwerp, 2610 Antwerp, Belgium; 9Laboratory of Medical Microbiology, Vaccine & Infectious Disease Institute, University of Antwerp, 2610 Antwerp, Belgium; 10Institut Universitari d’Investigació en Atenció Primària Jordi Gol (IDIAP Jordi Gol), 08007 Barcelona, Spain; 11Centro de investigación Biomédica en Red Enfermedades Infecciosas (CIBERINFEC), 28029 Madrid, Spain; 12Centre for Family and Community Medicine, Faculty of Health Sciences, Medical University of Lodz, 92-213 Lodz, Poland; 13National Center for Disease Control and Public Health, Tbilisi and Arner Science Management LLC, 0190 Tbilisi, Georgia; 14University Clinic of Primary Medical Assistance, State University of Medicine and Pharmacy “Nicolae Testemițanu”, MD-2004 Chişinǎu, Moldova; 15Julius Center for Health Sciences and Primary Care, University Medical Center Utrecht, Universiteitsweg 100, 3584 CG Utrecht, The Netherlands; 16Nuffield Department of Primary Care Health Sciences, University of Oxford, Oxford OX1 4BH, UK

**Keywords:** audit, primary health care, quality in healthcare, respiratory infections, antibiotic prescribing

## Abstract

Up to 80% of antibiotics are prescribed in the community. An assessment of prescribing by indication will help to identify areas where improvement can be made. A point prevalence audit study (PPAS) of consecutive respiratory tract infection (RTI) consultations in general practices in 13 European countries was conducted in January–February 2020 (PPAS-1) and again in 2022 (PPAS-4). The European Surveillance of Antibiotic Consumption quality indicators (ESAC-QI) were calculated to identify where improvements can be made. A total of 3618 consultations were recorded for PPAS-1 and 2655 in PPAS-4. Bacterial aetiology was suspected in 26% (PPAS-1) and 12% (PPAS-4), and an antibiotic was prescribed in 30% (PPAS-1) and 16% (PPAS-4) of consultations. The percentage of adult patients with bronchitis who receive an antibiotic should, according to the ESAC-QI, not exceed 30%, which was not met by participating practices in any country except Denmark and Spain. For patients (≥1) with acute upper RTI, less than 20% should be prescribed an antibiotic, which was achieved by general practices in most countries, except Ireland (both PPAS), Croatia (PPAS-1), and Greece (PPAS-4) where prescribing for acute or chronic sinusitis (0–20%) was also exceeded. For pneumonia in adults, prescribing is acceptable for 90–100%, and this is lower in most countries. Prescribing for tonsillitis (≥1) exceeded the ESAC-QI (0–20%) in all countries and was 69% (PPAS-1) and 75% (PPAS-4). In conclusion, ESAC-QI applied to PPAS outcomes allows us to evaluate appropriate antibiotic prescribing by indication and benchmark general practices and countries.

## 1. Introduction

In January 2022, the first joint report on antimicrobial resistance surveillance from the European Centre for Disease Prevention and Control (ECDC) and World Health Organization (WHO) Regional Office was published and provided a pan-European overview of the extent of antibiotic resistance (ABR) in Europe [1]. ABR is driven by and associated with antibiotic use. The European Surveillance of Antibiotic Consumption Network (ESAC-Net) reports on antibiotic consumption in the community with data from 30 EU/EEA countries [2]. Overall, Greece had the highest, while the Netherlands showed the lowest consumption relative to their population. However, to be able to evaluate the quality of prescribing, the prescription of antibiotics should be considered in the context of the infection for which they are prescribed. Few countries routinely record both prescribing and indication for antibiotics. The indications for appropriate antibiotic use for acute upper respiratory tract infections (URTI) depend largely upon the origin of the infection [3,4]. For instance, viruses cause the majority of URTIs, and antibiotics are not indicated [3,4,5]. However, for infections caused by bacteria, such as pertussis and acute bacterial sinusitis, antibiotics are indicated [3,6]. 

In 2007, ESAC published a set of 12 relevant evidence-based disease-specific quality indicators (QI) for outpatient antibiotic prescribing [7]. These QI can be applied in all European countries using a similar methodology but generally the indication for prescribing is lacking [8,9,10,11]. A retrospective observational database study in Belgium, Netherlands, and Sweden, where prescribing and indication were taken into account, showed wide variation between countries, which helped to adjust interventions and improve guidelines [9].

In 2020, a prospective point prevalence audit survey (PPAS) of consultation and management characteristics of patients presenting to their general practitioner (GP) with a respiratory tract infection (RTI) in 16 European countries was initiated [12]. In January and February 2020 (PPAS1), participating general practices were requested to anonymously register consultations of patients with RTI symptoms. This was repeated immediately after the start of the COVID pandemic (April/May 2020) as well as every year since (2021, 2022). The collected data provided the opportunity to compare and evaluate the antibiotic prescribing quality using ESAC QI between and within participating general practices in 13 European countries and identify opportunities for antibiotic stewardship. Participating general practices are referred to by the country where they are based, for ease of reading.

## 2. Results

A total of 3,618 consultations were recorded for PPAS-1 (range 132–390) and 2655 (range 198–233) in PPAS-4 (Table 1). The mean age in PPAS-1 was 35 (SD 25) and PPAS-4 was 36 (SD 23). Some differences in mean age were observed between PPAS-1 and PPAS-4 for in Belgium, Georgia, and Moldova, but overall, no change was observed. In PPAS-1, 31% of patients reported a comorbidity, which was 34% in PPAS-4. The range of comorbidities between countries was large and showed a range from 10% (Georgia, PPAS-1) to 50% (Greece, PPAS-4) (Table 1). Suspected bacterial aetiology dropped from 26% in PPAS-1 to 12% in PPAS-4. This decrease was observed in most countries, except for the UK and Germany. Overall, about 4% of consultations in PPAS-4 resulted in a hospital referral, ranging from 1% (Belgium) to 9% (Greece).

Table 2 shows the symptoms recorded in PPAS-1 and PPAS-4 by country. Cough was the most often reported symptom for consultation, with 84% (range 69–91%) of consultations in PPAS-1 and 66% (range 43% to 82%) in PPAS-4. Rhinitis was reported in 49% and 52% of consultations in PPAS-1 and PPAS-4, respectively, ranging from 25% (Spain) to 84% (Germany) in PPAS-1 and 28% (Spain) to 85% (Moldova) in PPAS-4. Sore throat was reported in 60% of consultations (35% Netherlands to 83% Moldova and Romania) in PPAS-1 and 52% (26% Netherlands to 94% Moldova) in PPAS-4.

Antibiotics were prescribed in 30% of consultations in PPAS-1 and 16% in PPAS-4 with wide variations between countries. The lowest prescribers were Belgium, Denmark, and Croatia in both PPAS-1 and PPAS-4 (Table 2 and Figure 1).

When prescribing for RTI, overall about 30% is prescribed a broad-spectrum penicillin, 15% a narrow-spectrum penicillin, 21% a macrolide, 19% amoxicillin-clavulanic acid, and quinolones and tetracyclines are prescribed around 5% each (Figure 2). This differs little between PPAS-1 and PPAS-4. However, a wide variety can be observed between countries, with GP practices in Denmark showing the most narrow-spectrum penicillin prescribing. Belgium, Netherlands, and UK all prescribe penicillin in more than 60% of the consultations, both broad and narrow spectrum. While other countries show preferences for other antibiotics, such as macrolides as well as amoxicillin-clavulanic acid, for a third to half of RTI consultations in Greece, Georgia, Croatia, Moldova, Poland, and Romania.

ESAC QI values were calculated for each country and for the overall total. Table 3 and Figure 3 present the overview for all the ESAC QI for each PPAS. Appendix A provides an overview of each ESAC QI as well as sub-indicators (i.e., those in the main indicator category who were prescribed antibiotics according to the QI) by country.

The percentage of adult patients with bronchitis who receive an antibiotic should, according to the QI, not exceed 30%. This QI was not met by any country, except for Denmark and Spain, in both PPASs. For patients (over one year of age) with acute upper RTI, less than 20% should be prescribed an antibiotic, which was achieved by most countries, except Ireland in both PPAS, Croatia (PPAS-1), and Greece (PPAS-4). In the same countries, antibiotic prescribing also exceeded the acceptable range of 0–20% for acute or chronic sinusitis. For pneumonia in adults, prescribing is acceptable in 90–100% of the cases, and this is lower in many countries. However, the number of patients in each country with pneumonia was relatively low (between 1 and 26). Prescribing for tonsillitis in the over-one-year age group exceeded the acceptable range of 0–20% for all countries and was 69% and 75% for PPAS-1 and PPAS-4, respectively (Figure 3).

An increase in prescribing for adult patients with acute bronchitis was observed between PPAS-1 and -4 while antibiotic prescribing for patients with acute upper RTI, cold, or sinusitis in both groups (over one year of age and adults) fell. For acute bronchitis in adults, this increase was mainly due to an increase in Georgia, Romania, the UK, and to a lesser extent the Netherlands, while Germany, Denmark, Croatia, and Ireland showed a decrease. The decrease in prescribing for acute upper RTI, cold, or sinusitis was mainly due to decreases in Croatia, Romania, Denmark, and Spain. The UK showed increases in the over-one-year and adult age groups for acute upper RTI, cold, or sinusitis. 

Quinolone prescribing should not exceed 5% as the antibiotic of choice for bronchitis. Only in Greece (PPAS-1 and -4) and Romania (PPAS-1) was this exceeded. For two consultations (Ireland and Romania), quinolones were prescribed for acute upper RTI in PPAS-1 and for five consultations (Germany, Greece, and Poland) in PPAS-4. Quinolones were occasionally prescribed for sinusitis (Romania, Germany, and Greece) and tonsillitis (Romania). For pneumonia, quinolones were prescribed in Spain (3), Greece (4), and the Netherlands (1) in PPAS-1, while in PPAS-4, one quinolone prescription was issued during a Greek and a Polish consultation.

## 3. Discussion

PPAS offers general practices in countries and a unified protocol to record RTI consultations, which allows comparisons of participating general practices between and within countries. The percentage of consultations with suspected bacterial aetiology dropped and so did the percentage of antibiotics prescribing. If there was prescribing, nearly half (45%) of the prescriptions were for a penicillin, but macrolide, amoxicillin-clavulanic acid, and quinolones remain popular choices. QI in relation to bronchitis and tonsillitis are generally not met, but overall better results were observed for acute upper RTI, sinusitis, and pneumonia.

There do seem to be differences between participating general practice countries in the type of patients consulting with up to 50% of patients with comorbidities in some countries. The differences in the management of patients as well as antibiotic prescribing may be explained by differences in patient consulting behaviour between countries [13]. Patients may present less often (self-manage), later, when comorbidities are present, or only if symptoms are more severe, and thereby have a higher likelihood of receiving an antibiotic [14,15]. The Netherlands, for instance, shows consistent lower antibiotic prescribing [2], while in this analysis, a relatively high percentage of prescribing was observed. This may be explained by the higher percentage of high-risk patients (higher co-morbidity) and a higher threshold to consultation with a GP (triage), resulting in more severe cases to be seen by the GP [10,16]. Further, in-depth and updated analysis of patient consultation behaviour between participating general practices and countries is planned [17]. 

More consultations were considered of bacterial aetiology in PPAS-1 (26%) compared to PPAS-4 (12%), also reflected in the proportion of antibiotic prescriptions (30% v 16%). Symptoms recorded during PPAS-1 and PPAS-4 were broadly similar within participating general practices in countries. Whether patients and GPs were more accepting of a viral diagnosis (and thus of not prescribing antibiotics) since the COVID-19 pandemic cannot be concluded from the results but may be a positive unintended consequence of the pandemic. A decrease in antibiotic prescribing since the pandemic has previously been described but did not persist [18]. Results from the next PPAS will allow confirmation of a lasting effect of the pandemic on antibiotic prescribing. 

In general, when the decision to prescribe an antibiotics is made, penicillin is recommended, in a particular narrow spectrum (J01CE). In many countries GPs still prescribe antibiotics, which are not recommended, such as macrolides and amoxicillin-clavulanic acid. This is generally due to non-adherence to guidelines and, in some countries, also due to availability of (narrow-spectrum) penicillin (personal communication within network). Education on the use of narrow-spectrum antibiotics could be considered for campaigns to improve the choice of antibiotic. Macrolides remain commonly prescribed for the treatment of RTIs, but the development of widespread macrolide resistance has limited their use [19]. In addition, macrolide resistance persists much longer compared to amoxicillin resistance, which is another reason to encourage amoxicillin prescribing, if prescribing is indicated [20,21]. Similarly, even though amoxicillin-clavulanic acid is commonly prescribed for RTI, due to its popularity and rising resistance, many countries now discourage the use of amoxicillin-clavulanic acid [22,23,24]. The high prescribing rate of macrolides and amoxicillin-clavulanic acid for a third to half of RTI consultations in GPs in six countries, where a high AMR is reported, indicates areas of high priorities for interventions. Quinolone prescribing is generally discouraged for respiratory infections [25], and quinolone prescribing is part of the QI. Most countries only see occasional quinolone use, but Greece has slightly more. This is in similar to the ECDC-ESAC reporting for the consumption of antibacterials for systemic use in the community in Europe (reporting year 2020), which showed that Greece was one of the highest prescribers of quinolones, which could be an additional focus area for this country [26].

When considering opportunities for antibiotic stewardship in general practice, targeted actions would have the most impact on antibiotic prescribing for tonsillitis. Around 70% of tonsillitis consultations result in an antibiotic prescription, when the acceptable range is between 0–20%. Much improvement can be made. Similarly, most countries could also improve the quality of antibiotic prescribing for bronchitis, with antibiotics prescribed during half of the consultations where the acceptable range is 0–30%. In particular, Ireland shows an excessive prescribing of antibiotics for upper RTI and sinusitis, and additional interventions should aim to reduce prescribing for these conditions, which is possible considering that other countries do not exceed the acceptable range. 

Differences between PPAS-1 and -4, overall as well as within participating general practices in countries, show that the ESAC QI are sensitive enough to pick up changes. Future PPASs together with national and international antibiotic strategies (ABS) will show if and where campaigns are working and may also provide evidence of what works, if countries use different styles of interventions and awareness campaigns. Interventions targeted at GP have shown successes and updates or adjustments to focus these campaigns in the light of our findings may be timely [27,28,29]. A recent review of public awareness campaigns on antibiotic prescribing and awareness highlighted that outcome measures to evaluate these campaigns differ widely, and no long-term outcomes (beyond 6 months) were included [30]. Additionally, it showed that social media was not used as a tool to increase public awareness, which may offer opportunities. Additional educational awareness campaigns to influence (the timing of) a patient’s consultation behaviour should also be considered. 

The main strength was the existence of the PPAS infrastructure and yearly monitoring of RTI in general practice with a pre-defined (standardised) set of questions. Most participating practices have multiple GPs contributing to the diversity in diagnosis and prescribing. This allows for prospective data collection and direct comparisons between participating general practices in countries, in particular when assessing country-specific guidelines. The recording of both antibiotic prescribing as well as its indication allows direct comparisons of the quality of prescribing between and within countries. The main limitation also lies in the existing infrastructure, which consisted of 2 to 10 general practices in each country, and even though multiple GPs may have contributed, the participating practices may not represent all general practices, nor is a comparison with other practices within each country possible. Participating practices may already have an interest in antibiotic prescribing and/or RTIs, which could result in better prescribing than could be expected with a larger sample of practices. However, as we can see changes over time within countries, the ESAC QI are sensitive enough to pick up small changes. When looking at individual countries and further breaking it down into the different QIs, the number of consultations decrease, and small changes or occasional prescribing seem to have a large effect. As PPAS is repeated yearly, timely publications of its results may encourage more countries, general practices, and GPs to participate and add to the generalisability of its results.

In conclusion, ESAC QI provide a high-level snapshot, which, when applied to PPAS outcomes, benchmark participating general practices in countries against each other. Even though differences remain, the similarities in poor outcomes (for tonsillitis and bronchitis) identified clear targets for antibiotic stewardship.

## 4. Materials and Methods

A prospective PPAS of consultation and management characteristics of patients with RTIs in 19 European countries was conducted using a pre-defined set of questions (Appendix A). GPs anonymously registered consultations of patients with RTI symptoms to describe overall patient care. PPAS-1 (January/February 2020), PPAS-2 (April/May 2020), PPAS-3 (January/February 2021), and PPAS-4 (January/February 2022) have been completed. The presented analysis focuses on PPAS-1 and -4 to avoid the pandemic impact, which has been described previously [12]. Methods and results of this study have been described in detail elsewhere [12].

### 4.1. Setting

Participating general practices in countries involved in the repeated PPAS and from which a full dataset was available for PPAS-1 and -4 were Belgium, Croatia, Denmark, Georgia, Germany, Greece, Ireland, Moldova, the Netherlands, Poland, Romania, Spain, and the United Kingdom (UK) (Figure 4). Armenia was excluded as they only recruited in a paediatric setting, while other countries recorded consultations in all age groups. Israel and Italy were excluded because they only collected data in a paediatric setting for PPAS-1 and only in long-term care for PPAS-4. Norway, Hungary, France, and Ukraine were excluded because they did not participate in PPAS-4. Nearly all consultations (86.4%) were in general practices that participated in both PPAS-1 and -4. 

As only consultation details were recorded and no personally identifiable information was collected, informed consent was not necessary. Each country obtained ethical approval or received a waiver for full ethical submission for the PPAS.

### 4.2. Eligibility Criteria

In PPAS-1 and -4, GPs were asked to sequentially register consultations of patients of all ages with either sore throat (symptom duration < 14 days) and/or cough (symptom duration < 28 days) and to exclude consultations of patients with only nasal or ear symptoms [12]. PPAS-4 patients with symptoms suggestive of COVID-19 were also included. Consultations for symptoms that were determined to be of non-infective origin or allergic symptoms were not included.

### 4.3. Data Analysis

Free responses for working diagnosis were analysed and coded according to the ESAC QI. The indicators are applied to consultations with patients within certain age ranges and with specific diagnoses and provide an acceptable percentage range of patients who are prescribed an antibiotic [8]. The ESAC QI apply the International Classifications for Primary Care (ICPC-2) to classify diagnoses of patients consulting their GP and the ATC (anatomical therapeutic chemical) classification to classify antibiotics.

The PPASs were designed for a broader application than to evaluate the ESAC quality indicators and the following adjustments were made:oDue to the focus on respiratory symptoms and the exclusion of those with solely ear symptoms, for ESAC Quality Indicators 3 (focusing on urinary tract infections) and 6 (focusing on otitis media) values could not be calculated.oESAC QI for upper RTI and sinusitis were combined with the working diagnosis “cold,” as this is how it was presented on the PPAS-4 survey form (Appendix A). For the analysis of the QI, the combined diagnosis was used for both QIs [8].oTable 4 shows the antibiotic prescribing options on the PPAS survey and its allocated to ESAC Quality Indicators.

IBM SPSS (Version 27) and R (version 4.1.0) were used to conduct the analysis and generate graphs.

## Figures and Tables

**Figure 1 antibiotics-12-00572-f001:**
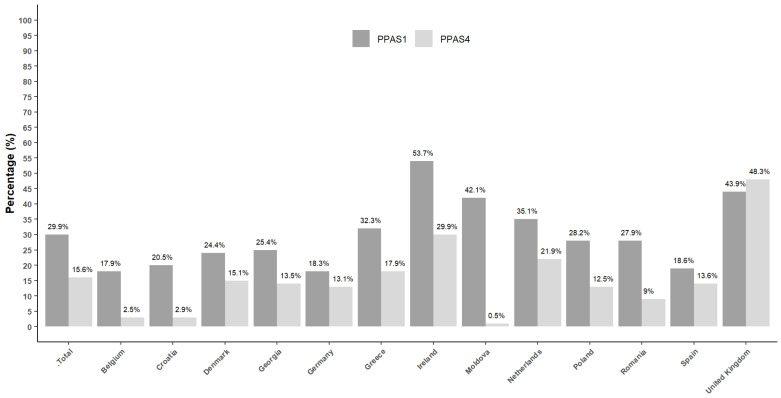
Overview of the percentage of consultations resulting in an antibiotic prescription, overall and by GPs in each country in PPAS-1 and PPAS-4.

**Figure 2 antibiotics-12-00572-f002:**
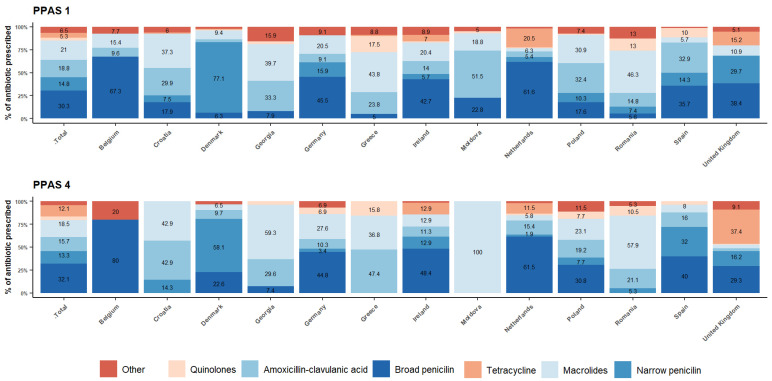
Overview of the percentage of each antibiotic prescribed, overall and by country in PPAS-1 and PPAS-4.

**Figure 3 antibiotics-12-00572-f003:**
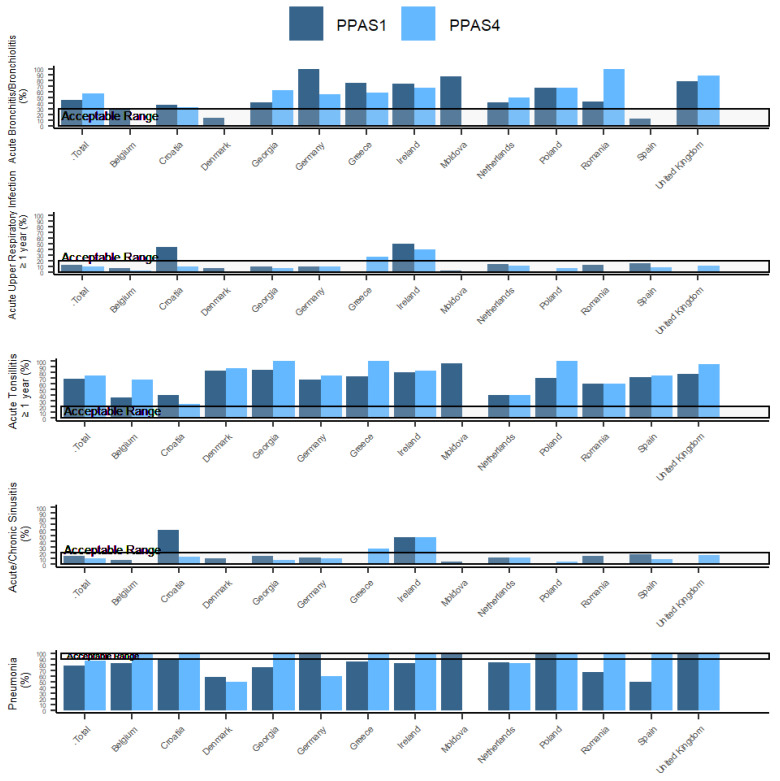
ESAC quality indicators (acceptable range) for PPAS-1 and PPAS-4. Percentage antibiotic prescribing overall and by country, for each condition for adults, unless otherwise stated.

**Figure 4 antibiotics-12-00572-f004:**
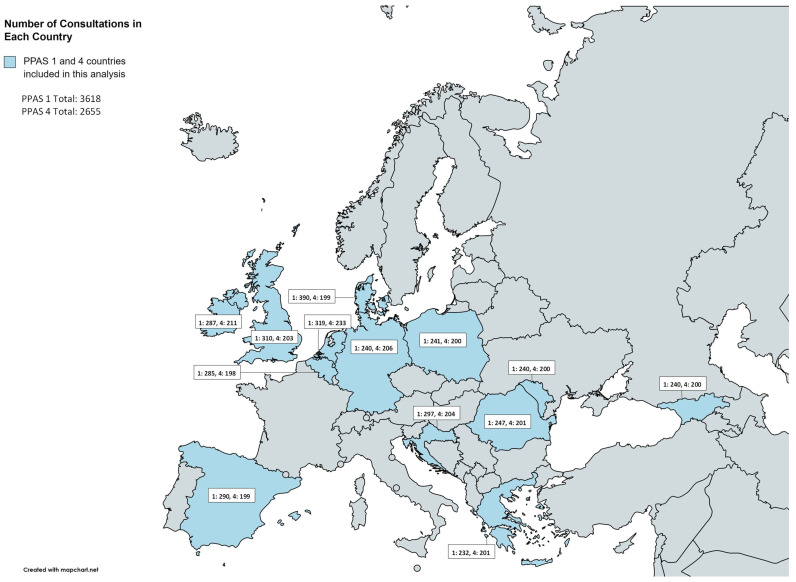
Number of consultations per participating country in PPAS-1 and -4.

**Table 1 antibiotics-12-00572-t001:** Characteristics of recorded consultations in PPAS-1 and PPAS-4 by country.

	Total Consultations Recorded		Age (Years)				Comorbidity				Suspected Bacterial Aetiology *				Hospitalisation			
	PPAS-1	PPAS-4	PPAS-1		PPAS-4		PPAS-1		PPAS-4		PPAS-1		PPAS-4		PPAS-1		PPAS-4	
	N	N	Mean	SD	Mean	SD	n	%	n	%	n	%	n	%	n	%	n	%
Belgium	285	198	37.7	25.0	28.6	18.3	78	27.4%	25	12.6%	46	16.1%	4	2.0%	5	1.8%	2	1.0%
Croatia	297	204	36.7	27.7	39.1	21.4	76	25.6%	68	33.3%	51	17.2%	2	1.0%	13	4.4%	5	2.5%
Denmark	390	199	28.8	23.4	32.5	27.3	78	20.0%	75	37.7%	96	24.6%	34	17.1%	13	3.3%	12	6.0%
Georgia	240	200	25.1	19.3	41.0	19.1	23	9.6%	71	35.5%	42	17.5%	3	1.5%	2	0.8%	7	3.5%
Germany	240	206	31.9	20.9	39.2	16.1	89	37.1%	71	34.5%	43	17.9%	36	17.5%	1	0.4%	3	1.5%
Greece	232	201	48.2	21.3	45.7	20.7	94	40.5%	100	49.8%	66	28.4%	26	12.9%	9	3.9%	17	8.5%
Ireland	287	211	25.8	22.9	25.6	23.6	133	46.3%	57	27.0%	98	34.1%	36	17.1%	10	3.5%	8	3.8%
Moldova	240	200	47.8	25.0	37.5	20.2	55	22.9%	76	38.0%	100	41.7%	6	3.0%	4	1.7%	10	5.0%
Netherlands	319	233	26.8	23.9	34.9	28.2	147	46.1%	89	38.2%	93	29.2%	41	17.6%	7	2.2%	9	3.9%
Poland	241	200	27.2	22.0	28.5	23.7	66	27.4%	64	32.0%	61	25.3%	25	12.5%	5	2.1%	6	3.0%
Romania	247	201	49.0	23.2	35.4	20.6	68	27.5%	69	34.3%	77	31.2%	14	7.0%	2	0.8%	13	6.5%
Spain	290	199	38.7	17.6	41.0	19.7	116	40.0%	66	33.2%	52	17.9%	22	11.1%	5	1.7%	5	2.5%
UK	310	203	30.9	23.4	34.2	26.3	79	25.5%	63	31.0%	107	34.5%	80	39.4%	7	2.3%	2	1.0%
Total	3618	2655	35.1	24.5	35.6	22.9	1102	30.5%	894	33.7	932	25.8%	329	12.4%	83	2.3%	99	3.7%

* According to GP’s impression.

**Table 2 antibiotics-12-00572-t002:** Overview of symptoms and antibiotic prescribing recorded in PPAS-1 and PPAS-4 by country.

	Rhinitis *				Sore Throat *				Cough *				Antibiotics			
	PPAS-1		PPAS-4		PPAS-1		PPAS-4		PPAS-1		PPAS-4		PPAS-1		PPAS-4	
	n	%	n	%	n	%	n	%	n	%	n	%	n	%	n	%
Belgium	169	60.1%	136	69.0%	193	70.7%	113	57.9%	245	86.6%	142	72.4%	51	17.9	5	2.5
Croatia	112	37.7%	75	36.8%	190	64.0%	68	33.5%	249	83.8%	97	47.5%	61	20.5	6	2.9
Denmark	211	58.1%	84	48.3%	171	51.0%	77	54.2%	323	84.6%	138	70.4%	95	24.4	30	15.1
Georgia	104	43.3%	60	30.0%	159	66.3%	108	54.0%	218	90.8%	115	57.8%	61	25.4	27	13.5
Germany	198	83.9%	131	63.9%	142	60.7%	136	68.3%	189	80.1%	116	58.3%	44	18.3	27	13.1
Greece	152	66.1%	86	43.7%	161	69.7%	92	45.8%	212	91.4%	149	74.1%	75	32.3	36	17.9
Ireland	108	37.9%	93	47.7%	106	37.9%	89	45.2%	252	88.1%	154	74.0%	154	53.7	63	29.9
Moldova	123	51.2%	170	85.0%	197	82.8%	186	93.5%	177	73.8%	86	43.0%	101	42.1	1	0.5
Netherlands	178	59.9%	121	53.1%	101	34.7%	57	25.7%	283	89.6%	175	75.8%	112	35.1	51	21.9
Poland	123	51.2%	134	67.3%	146	62.1%	78	39.8%	216	89.6%	148	74.4%	68	28.2	25	12.5
Romania	114	47.5%	140	73.3%	188	82.8%	153	80.5%	201	82.0%	132	66.0%	69	27.9	18	9
Spain	72	24.8%	56	28.1%	135	46.6%	87	43.9%	239	82.4%	108	54.3%	54	18.6	27	13.6
UK	75	25.2%	59	30.1%	184	60.5%	78	41.1%	212	68.8%	166	82.2%	136	43.9	98	48.3
Total	1739	49.2	1354	52.0	2073	59.7	1322	52.2	3016	83.9	1726	65.5	1081	29.9	414	15.6

* Tick box answers (see Appendix A). Unknown recorded to no. Patients may have had more than one symptom.

**Table 3 antibiotics-12-00572-t003:** ESAC Quality Indicators by country for PPAS-1 (2020) and -4 (2022). Highlighted are the percentages that exceed the acceptable range for each QI.

Country	Percentage of Patients Aged 18–75 Years with Acute Bronchitis/Bronchiolitis Prescribed Antibiotics (ESAC 1A)				Percentage of Patients ≥ 1 Year with Acute Upper Respiratory Infection Prescribed Antibiotics (ESAC 2A)				Percentage of Patients ≥ 1 Year with Acute Tonsillitis Prescribed Antibiotics (ESAC 4A)				Percentage of Patients ≥ 18 Years with Acute/Chronic Sinusitis Prescribed Antibiotics (ESAC 5A)				Percentage of Patients Aged 18–65 Years with Pneumonia Prescribed Antibiotics (ESAC 7A)			
	Acceptable Range 0–30%				Acceptable Range 0–20%				Acceptable Range 0–20%				Acceptable Range 0–20%				Acceptable Range 90–100%			
	PPAS-1		PPAS-4		PPAS-1		PPAS-4		PPAS-1		PPAS-4		PPAS-1		PPAS-4		PPAS-1		PPAS-4	
	n	%	n	%	n	%	n	%	n	%	n	%	n	%	n	%	n	%	n	%
Belgium	9	31	-	-	5	7	1	2	16	36	2	67	4	8	1	2	9	82	1	100
Croatia	10	37	1	33	4	44	2	9	6	40	1	25	3	60	2	13	8	89	2	100
Denmark	7	14	0	0	2	7	0	0	32	84	13	87	2	10	0	0	16	59	3	50
Georgia	22	42	10	63	1	9	4	7	11	85	4	100	1	14	3	7	3	75	9	100
Germany	10	100	6	55	6	10	12	10	8	67	3	75	6	11	12	10	4	100	6	60
Greece	30	75	11	58	-	-	5	26	8	73	5	100	-	-	5	28	6	86	3	100
Ireland	26	74	4	67	24	49	22	40	21	81	10	83	19	48	14	47	9	82	3	100
Moldova	13	87	-	-	2	3	0	0	45	96	0	0	1	4	0	0	1	100	-	-
Netherlands	13	42	3	50	8	14	11	11	7	41	4	40	5	11	6	11	26	84	5	83
Poland	8	67	4	67	0	0	6	7	10	71	2	100	0	0	2	5	6	100	5	100
Romania	12	43	4	100	2	13	0	0	38	61	6	60	2	15	0	0	4	67	2	100
Spain	9	13	0	0	2	15	5	8	13	72	12	75	2	17	5	9	3	50	2	100
UK	28	78	21	88	0	0	9	11	53	77	18	95	0	0	7	16	7	100	11	100
Total	197	45	64	57	56	13	77	10	268	69	80	76	45	15	57	10	102	79	52	87

Calculation of percentages: n (numerator)/N (denominator-total of patients with the diagnosis; see Appendix A) × 100.

**Table 4 antibiotics-12-00572-t004:** Allocation of antibiotic prescribing from PPAS survey options to ESAC Quality Indicators.

ESAC Quality Indicators	Options of Antibiotic Prescribing in PPAS Survey
ESAC 1A: Percentage of patients aged 18–75 years with acute bronchitis/bronchiolitis prescribed antibiotics (ATC: J01) ESAC 2A: Percentage of patients older than 1 year with acute upper respiratory infection, cold, or sinusitis, prescribed antibacterials (ATC: J01) ESAC 4A: Percentage of patients older than 1 year with acute tonsillitis prescribed antibacterials (ATC: J01) ESAC 5A: Percentage of patients older than 18 years with sinusitis, upper respiratory tract infection, or cold prescribed an antibacterial (ATC: J01) ESAC 7A: Percentage of patients aged between 18 and 65 prescribed antibacterials (ATC: J01)	Tetracycline (e.g., doxycycline) (ATC: J01AA) Narrow-spectrum penicillin (e.g., calvapen, phenoxymethyl penicillin) (ATC: J01CE) Broad-spectrum penicillin (e.g., amoxicillin) (ATC: J01CA) Co-amoxiclav (amoxicillin-clavulanate acid) (ATC: J01CR) Macrolide (e.g., clarithromycin, erythromycin) (ATC: J01F) Quinolone (e.g., ciprofloxacin, levofloxacin) (ATC: J01M) Cephalosporin (e.g., cephalexin) (ATC: J01D) Other
ESAC 1B_1: 1A receiving the recommended antibacterials: (ATC: J01CA * or J01AA) ESAC 7B_1: 7A receiving the recommended antibacterials (ATC: J01CA * or J01AA)	Tetracycline (e.g., doxycycline) (ATC: J01AA) Broad-spectrum penicillin (e.g., amoxicillin) (ATC: J01CA)
ESAC 1B_2: 1A receiving the recommended antibacterials (ATC: J01CA * or J01CE * or J01AA) ESAC 7B_2: 7A receiving the recommended antibacterials (ATC: J01CA * or J01CE * or J01AA)	Tetracycline (e.g., doxycycline) (ATC: J01AA) Narrow-spectrum penicillin (e.g., calvapen, phenoxymethyl penicillin) (ATC: J01CE) Broad-spectrum penicillin (e.g., amoxicillin) (ATC: J01CA)
ESAC 1C: 1A receiving quinolones (ATC: J01M) ESAC 2C: 2A receiving quinolones (ATC: J01M) ESAC 4C: 4A receiving quinolones (ATC: J01M) ESAC 5C: 5A receiving quinolones (ATC: J01M) ESAC 7C: 7A receiving quinolones (ATC: J01M)	Quinolone (e.g., ciprofloxacin, levofloxacin) (ATC: J01M)
ESAC 2B_1: 2A receiving the recommended antibacterials (ATC: J01CE) * ESAC 4B_1: 4A receiving the recommended antibacterials (ATC: J01CE) *	Narrow-spectrum penicillin (e.g., calvapen, phenoxymethyl penicillin) (ATC: J01CE)
ESAC 2B_2: 2A receiving the recommended antibacterials (ATC: J01CA or J01CE) * ESAC 4B_2: 4A receiving the recommended antibacterials: (ATC: J01CA or J01CE) * ESAC 5B: 5A receiving the recommended antibacterials (ATC: J01CA or J01CE) *	Narrow-spectrum penicillin (e.g., calvapen, phenoxymethyl penicillin) (ATC: J01CE) Broad-spectrum penicillin (e.g., amoxicillin) (ATC: J01CA)

* The options on the PPAS survey for penicillin that were prescribed were either “broad-spectrum penicillin” or “narrow-spectrum penicillin.” The penicillin classes in the ATC, used in the ESAC indicators, were J01CA (penicillin with extended spectrum) and J01CE (beta-lactamase sensitive penicillin). To compensate for this, ESAC indicators, including any type of penicillin as the recommended antibiotic, were analysed twice, once with broad-spectrum penicillin treated as J01CA and narrow-spectrum penicillin treated as J01CE and once where narrow- and broad-spectrum penicillin were combined into one category, which was applied to both J01CA and J01CE.

## Data Availability

Not applicable.

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
