# Peer review of "Disease-Specific Quality Indicators for Outpatient Antibiotic Prescribing for Respiratory Infections (ESAC Quality Indicators) Applied to Point Prevalence Audit Surveys in General Practices in 13 European Countries"

_antibiotics, 2023, doi:10.3390/antibiotics12030572_

Round 1
Reviewer 1 Report
In their manuscript, “European wide benchmarking of disease-specific quality indicators for outpatient antibiotic prescribing for respiratory infections: ESAC quality indicators applied to point prevalence audit surveys in 13 European countries,” Vellinga and colleagues report their two-time – Jan/Feb 2020 and Jan/Feb 2022 – cross sectional convenience sample of visits by patients with respiratory symptoms to practices throughout Europe and adherence to European Surveillance of Antibiotic Consumption (ESAC) quality indicators. According to an earlier publication (BMJ Open 2021), in Jan/Feb 2020, participating GPs were asked to sequentially register consulting patients who had sore throat for fewer than 14 days or cough for fewer than 28 days, and, after April/May 2020, also register patients with coryza or for whom they otherwise suspected COVID-19.
The study included 3618 consultations in Jan/Feb 2020 and 2655 consultations in Jan/Feb 2022. The authors noted differences in the proportion of patients with suspected bacterial etiology and antibiotic prescribing rates for respiratory tract infection quality indicators between the time periods and among countries.
Although the data are interesting, it seems the data are not representative and it is a mischaracterization to say these findings are “European wide” or, as stated in the first sentence of the Discussion, “…which allows comparisons between and within countries.” The language in the previously cited publication (BMJ Open 2021) and the present manuscript refer to “national networks,” “countries,” and specific country names. However, the consultations appear to come from a convenience sample of GPs and visits. It is not entirely clear that the same GPs, clinics, or type of patients are involved between Jan/Feb 2020 and Jan/Feb 2022. Many of the patient characteristics within countries shifted dramatically from 2020 to 2022. Who are the participating clinics? GPs? Is it the authors? Someone else?
Related, the sample size is actually quite small from which to draw continental conclusions: 2655 consultations in 2022 spread across 13 countries and 5 quality measures gives an average denominator of 41 consultations per quality measure per country. With some antibiotic prescribing rates of less than 20%, these data are over stratified to make generalizable comparisons within and among countries.
In the first publication using these data, the investigators reported on a convenience sample of 4376 visit to 16 countries in Jan/Feb 2020 and 3301 patients in April/May 2020. This publication seems like a better use of the data: the earlier study described clinicians’ use of telehealth and confidence in treatment early in the pandemic. The current study is generalizing about international differences in antibiotic prescribing.
Specific Comments
Page 1, line 47: ESAC is not defined.
Page 3: The authors do not state how “suspected bacterial aetiology” was defined or operationalized.
Page 5: Table 2 has many more than the total number of patients sampled represented. Some explanation is warranted. The antibiotics column is redundant with Figure 1.
Page 8, Table 3: It is not stated if the Ns are the denominator or the numerator in the calculation of the antibiotic prescribing rate (I presume these are numerators).
Author Response
Point by Point Response to Reviewers:
Comments |
Response |
Reviewer #1: English language and style: (x) Moderate English changes required Must be improved: - Does the introduction provide sufficient background and include all relevant references? - Is the research design appropriate? - Are the methods adequately described? - Are the results clearly presented? - Are the conclusions supported by the results? In their manuscript, “European wide benchmarking of disease-specific quality indicators for outpatient antibiotic prescribing for respiratory infections: ESAC quality indicators applied to point prevalence audit surveys in 13 European countries,” Vellinga and colleagues report their two-time – Jan/Feb 2020 and Jan/Feb 2022 – cross sectional convenience sample of visits by patients with respiratory symptoms to practices throughout Europe and adherence to European Surveillance of Antibiotic Consumption (ESAC) quality indicators. According to an earlier publication (BMJ Open 2021), in Jan/Feb 2020, participating GPs were asked to sequentially register consulting patients who had sore throat for fewer than 14 days or cough for fewer than 28 days, and, after April/May 2020, also register patients with coryza or for whom they otherwise suspected COVID-19. |
We thank you for all the comments and suggestions that have helped to improve our manuscript. Native speaker English co-authors checked the English language and style of the manuscript. We added the suggestions of the reviewer to improve the manuscript. We changed follow reviewer’s comments accordingly:
We clarified the Eligibility Criteria, page 14, lines 294 to 301, which now reads: “In PPAS-1 and 4, GPs were asked to sequentially register consultations of patients of all ages with either sore throat (symptom duration<14 days) and/or cough (symptom duration<28 days) and to exclude consultations of patients with only nasal or ear symptoms [12]. In PPAS 4 patients with symptoms suggestive of COVID-19 were also included. Consultations for symptoms that were determined to be of non-infective origin or allergic symptoms were not included.” |
The study included 3618 consultations in Jan/Feb 2020 and 2655 consultations in Jan/Feb 2022. The authors noted differences in the proportion of patients with suspected bacterial etiology and antibiotic prescribing rates for respiratory tract infection quality indicators between the time periods and among countries. Although the data are interesting, it seems the data are not representative and it is a mischaracterization to say these findings are “European wide” or, as stated in the first sentence of the Discussion, “…which allows comparisons between and within countries.” The language in the previously cited publication (BMJ Open 2021) and the present manuscript refer to “national networks,” “countries,” and specific country names. However, the consultations appear to come from a convenience sample of GPs and visits. It is not entirely clear that the same GPs, clinics, or type of patients are involved between Jan/Feb 2020 and Jan/Feb 2022. Many of the patient characteristics within countries shifted dramatically from 2020 to 2022. Who are the participating clinics? GPs? Is it the authors? Someone else? |
We clarified the Eligibility Criteria, page 14, lines 293 to 297. Also, we clarified in Setting “Nearly all consultations (86.4%) were in general practices that participated in both PPAS 1 and 4. ” (page 13, Line 286)
We would argue that the data is European wide, as more than 3,600 in PPAS 1 and 2,655 consultations were recorded in 13 European countries. Allowing comparisons between these 13 countries. With more than 85% of the practices participating in both PPAS, a comparison of consultations can be made within countries as well. We do acknowledge in the limitations that the participating practices may not represent all practices and some practices may be more inclined to participate.
|
Related, the sample size is actually quite small from which to draw continental conclusions: 2655 consultations in 2022 spread across 13 countries and 5 quality measures gives an average denominator of 41 consultations per quality measure per country. With some antibiotic prescribing rates of less than 20%, these data are over stratified to make generalizable comparisons within and among countries. In the first publication using these data, the investigators reported on a convenience sample of 4376 visit to 16 countries in Jan/Feb 2020 and 3301 patients in April/May 2020. This publication seems like a better use of the data: the earlier study described clinicians’ use of telehealth and confidence in treatment early in the pandemic. The current study is generalizing about international differences in antibiotic prescribing. |
The sample size that is applicable for the quality measures is the actual number of consultations as these are descriptives and not statistical analyses performed. In case of statistical analysis, in particular regressions, the average denominator would apply, however, for descriptives, the denominator is the actual number. We acknowledge the drop in prescribing when stratification is done to compare prescriptions for each condition. This is why we provide both the percentages and the actual number, to allow context for interpretation.
PPAS data collected has served different analyses. The previous publication focused on the implications of the pandemic on consultations. In the presented analysis, we aimed to identify areas where improvements can be made, using standardised quality indicators. Each analysis has its own strengths and limitations. |
Page 1, line 47: ESAC is not defined.
Page 3: The authors do not state how “suspected bacterial aetiology” was defined or operationalized.
Page 5: Table 2 has many more than the total number of patients sampled represented. Some explanation is warranted. The antibiotics column is redundant with Figure 1.
Page 8, Table 3: It is not stated if the Ns are the denominator or the numerator in the calculation of the antibiotic prescribing rate (I presume these are numerators).
|
We thank the reviewer for these suggestions and changed accordingly: Page 1, line 47: “The European Surveillance of Antibiotic Consumption quality indicators (ESAC-QI)…”
Page 4: we added the definition as a footnote in table 1 “* According to GP’s impression”
Page 6: This may have been confusing for the reviewer, as the patient may have had more than one symptom. However, we have added a footnote (table 2) to clarify: "Ticked box answers (see Text S1 and S2). Unknown recoded to no. Patients may have had more than one symptom" We hope this will make it clearer. Also, we consider leaving the antibiotic column because figure 1 only shows the percentages. This also addresses the previous comment of this reviewer in relation to providing context for low numbers.
Page 9: We have clarified in table 3 how the percentages were calculated as a footnote “Calculation of percentages: n (numerator) / N (denominator - total of patients with the diagnosis, see table S2) * 100”. Also, we added a supplementary table with denominators of this calculation. |

Reviewer 2 Report
In this manuscript, the Authors describe their findings from a series of point prevalence audits of consecutive respiratory tract infection consultations across 13 European countries. Overall, they found high levels of inappropriate antibiotic prescribing across most European countries surveyed, particularly for acute bronchitis in adults and acute tonsilitis.
I have a few suggestions that I think may strengthen this manuscript:
Major Suggestions:
1. It would be helpful if the Authors discuss the microbiology of acute upper respiratory tract infections in the Introduction section. Many readers will be aware that the majority of acute upper respiratory tract infections are caused by viruses and that antibiotics are not indicated for these infections, but not all.
2. In addition, the Authors could briefly mention that antibiotics are only rarely indicated for some acute respiratory tract infections, such as pertussis and acute bacterial sinusitis (in some cases). A good review on acute sinusitis and when to prescribe antibiotics is here: Rosenfeld RM. CLINICAL PRACTICE. Acute Sinusitis in Adults. N Engl J Med. 2016 Sep 8;375(10):962-70. doi: 10.1056/NEJMcp1601749.
Minor Suggestion:
1. I may have missed it, but I don't see any mention in the "Methods" section regarding IRB approval of this study. This should be added.
2. If the Authors could reformat Table 3 so the full name of the country is on 1 line only, I think the Table would be more easily readable.
Author Response
Point by Point Response to Reviewers:
Comments |
Response |
Reviewer #2: English language and style: (x) English language and style are fine/minor spell check required Can be improved: - Does the introduction provide sufficient background and include all relevant references? - Are the results clearly presented? |
We thank you for all the comments and suggestions that have helped to improve our manuscript. We added the suggestions of the reviewer to improve the manuscript. We changed follow reviewer’s comments accordingly:
|
In this manuscript, the Authors describe their findings from a series of point prevalence audits of consecutive respiratory tract infection consultations across 13 European countries. Overall, they found high levels of inappropriate antibiotic prescribing across most European countries surveyed, particularly for acute bronchitis in adults and acute tonsilitis.
I have a few suggestions that I think may strengthen this manuscript:
Major Suggestions:
1. It would be helpful if the Authors discuss the microbiology of acute upper respiratory tract infections in the Introduction section. Many readers will be aware that the majority of acute upper respiratory tract infections are caused by viruses and that antibiotics are not indicated for these infections, but not all.
2. In addition, the Authors could briefly mention that antibiotics are only rarely indicated for some acute respiratory tract infections, such as pertussis and acute bacterial sinusitis (in some cases). A good review on acute sinusitis and when to prescribe antibiotics is here: Rosenfeld RM. CLINICAL PRACTICE. Acute Sinusitis in Adults. N Engl J Med. 2016 Sep 8;375(10):962-70. doi: 10.1056/NEJMcp1601749. |
We added a sentence to the introduction section to the microbiology of acute upper respiratory tract infections (viral and bacterial infection) and added references. Page 2, lines 74 to 78, now reads: “The indications for appropriate antibiotic use for acute upper respiratory tract infections (URTI) depend largely upon the origin of the infection [3,4]. For instance, viruses cause the majority of URTIs and antibiotics are not indicated [3-5]. However, for infections caused by bacteria, such as pertussis and acute bacterial sinusitis, antibiotics are indicated [3,6].” |
Minor Suggestion:
1. I may have missed it, but I don't see any mention in the "Methods" section regarding IRB approval of this study. This should be added.
2. If the Authors could reformat Table 3 so the full name of the country is on 1 line only, I think the Table would be more easily readable. |
We thank the reviewer for these minor suggestions and changed accordingly: 1. Ethical approvals were included on page 14 line 291 “Each country obtained ethical approval or received a waiver for full ethical submission for the PPAS”. Also, it was mentioned in a specific section “Institutional Review Board Statement” (see page 16, line 340).
2. We oriented the page setup of table 3 (portrait to landscape) to see the full name of the country on 1 line. |

Reviewer 3 Report
1. What is the main question addressed by the research? The main question addressed by the research is antibiotic consumption in European countries. 2. Do you consider the topic original or relevant in the field? I consider this topic relevant in the field as the data for many countries is limited. 3. What does it add to the subject area compared with other published material? Compared to previously published materials, this article involves several countries and large sample. 4. What specific improvements should the authors consider regarding the methodology? I do not have any comments on the methodology, as it seems appropriate. 5. Are the conclusions consistent with the evidence and arguments presented and do they address the main question posed? Conclusions are consistent with the presented data. 6. Are the references appropriate? The references are appropriate. 7. Please include any additional comments on the tables and figures. Figure quality needs to be improved.
Line 64 - define WHO
Figure 1 - please separate proportions from columns so it is easier to follow
Figure 4 - please improve the quality of this figure
please improve English language
overall, this is very important manuscript which i believe is suitable for publication in this highly respected journal.
Author Response
Point by Point Response to Reviewers:
Comments |
Response |
Reviewer #3: English language and style: (x) Moderate English changes required |
We thank you for all the comments and suggestions that have helped to improve our manuscript. Native English speakers checked the English language and style of the manuscript. |
1. What is the main question addressed by the research? The main question addressed by the research is antibiotic consumption in European countries. 2. Do you consider the topic original or relevant in the field? I consider this topic relevant in the field as the data for many countries is limited. 3. What does it add to the subject area compared with other published material? Compared to previously published materials, this article involves several countries and large sample. 4. What specific improvements should the authors consider regarding the methodology? I do not have any comments on the methodology, as it seems appropriate. 5. Are the conclusions consistent with the evidence and arguments presented and do they address the main question posed? Conclusions are consistent with the presented data. 6. Are the references appropriate? The references are appropriate. 7. Please include any additional comments on the tables and figures. Figure quality needs to be improved.
Line 64 - define WHO
Figure 1 - please separate proportions from columns so it is easier to follow
Figure 4 - please improve the quality of this figure
please improve English language
overall, this is very important manuscript which i believe is suitable for publication in this highly respected journal. |
We have addressed each of the reviewer’s comments as follows:
Line 65: we added the acronym “…World Health Organisation (WHO)…”
Figure 1 – We separated proportions from columns (page 5)
Figure 4 – We improved the quality (page 13) |

Round 2
Reviewer 1 Report
I do not think the authors addressed two main criticisms substantively. 1. Who are these GPs and what were these practices? How were they selected? Are they the authors themselves? 2. The authors are continuing to make comments about international differences when the clinics are, as noted in the manuscript, non-representative. The language throughout the entire article -- starting with the title and throughout -- needs to be toned down reflect that these are differences between practices in different countries, not make generalizations about differences in antibiotic prescribing between country X and country Y.
Author Response
Please find attached the revised manuscript as well as point-by-point responses to the comments of the reviewer and the section of the manuscript where the changes can be found.
